# BBS Proteins Affect Ciliogenesis and Are Essential for Hedgehog Signaling, but Not for Formation of iPSC-Derived RPE-65 Expressing RPE-Like Cells

**DOI:** 10.3390/ijms22031345

**Published:** 2021-01-29

**Authors:** Caroline Amalie Brunbjerg Hey, Lasse Jonsgaard Larsen, Zeynep Tümer, Karen Brøndum-Nielsen, Karen Grønskov, Tina Duelund Hjortshøj, Lisbeth Birk Møller

**Affiliations:** Department of Clinical Genetics, Copenhagen University Hospital, Rigshospitalet, Gl. Landevej 7, 2600 Glostrup, Denmark; caroline.amalie.brunbjerg.hey@regionh.dk (C.A.B.H.); lasse.jonsgaard.larsen@regionh.dk (L.J.L.); zeynep.tumer@regionh.dk (Z.T.); Karen.Broendum-Nielsen@regionh.dk (K.B.-N.); Karen.Groenskov@regionh.dk (K.G.); Tina.Duelund.Hjortshoej@regionh.dk (T.D.H.)

**Keywords:** BBS, WNT, hedgehog-signaling, ciliogenesis, SMO, induced pluripotent stem cell, RPE, RPE65

## Abstract

Bardet-Biedl syndrome (BBS) is a ciliopathy characterized by retinal dystrophy, renal cysts, obesity and polydactyly. BBS genes have been implicated in ciliogenesis, hedgehog signaling and retinal pigment epithelium maturation. BBS1 and BBS5 are members of the BBSome, implicated in cilia transport of proteins, and BBS10 is a member of the chaperonin-complex, mediating BBSome assembly. In this study, involvement of BBS1, BBS5 and BBS10 in ciliogenesis and hedgehog signaling were investigated in BBS-defective patient fibroblasts as well as in RPE-hTERT cells following siRNA-mediated knockdown of the BBS genes. Furthermore, the ability of BBS1-defective induced pluripotent stem-cells (iPSCs) to differentiate into RPE cells was assessed. We report that cells lacking functional BBS5 or BBS10 have a reduced number of primary cilia, whereas cells lacking functional BBS1 display shorter primary cilia compared to wild-type cells. Hedgehog signaling was substantially impaired and Smoothened, a component of hedgehog signaling, was trapped inside the cilia of the BBS-defective cells, even in the absence of Smoothened agonist. Preliminary results demonstrated the ability of BBS1-defective iPSC to differentiate into RPE-65 expressing RPE-like cells. The BBS1^−/−^-defective RPE-like cells were less pigmented, compared to RPE-like cells differentiated from control iPSCs, indicating an impact of BBS1 on RPE maturation.

## 1. Introduction

Bardet-Biedl syndrome (BBS) is a rare autosomal recessive inherited ciliopathy with prevalence of about 1:160,000 in European populations [1]. To date, 23 genes have been associated with BBS. BBS is pleiotropic, characterized by a broad spectrum of symptoms, affecting multiple organ systems. The primary features are retinal photoreceptor degeneration, obesity, polydactyly, renal anomalies, genital abnormalities and intellectual disabilities. Except for polydactyly, most clinical features are absent at birth with onset later in life. Photoreceptor degeneration with early macular involvement is a predominant feature with symptoms appearing during the first or second decade of life [2,3]. 

Ciliopathies is a group of hereditary diseases characterized by primary cilium dysfunction. Primary cilia are microtubule-based membrane protrusions present in a single copy on cells in quiescence [1,3,4,5,6,7]. Primary cilia are immotile but share structural features with motile cilia. Both types are composed of 9 doublet microtubules in the shape of a ring forming the axoneme. The primary cilium is anchored to the cell by the basal body (BB), which develops from the mother centriole of the centrosome in a manner that is coordinated and regulated along with the cell cycle. Most cells begin to disassemble their primary cilia at cell cycle re-entry. Suppression of ciliogenesis is required for cells to proliferate. Centrioles have a double function, in that they can give rise to the centrosome or convert to BBs, forming the base for the cilia. Ciliogenesis per se is highly regulated by extracellular and intracellular signaling [8].

Primary cilia are considered as separate organelles as their membrane compositions differ from that of the cell they protrude from. Primary cilia function as sensory organelles that coordinate a variety of signal transduction pathways, such as Hedgehog- (Hh), transforming growth factor Beta (TGF-Beta)- and WNT signaling that are critical for regulating cell polarity, differentiation, migration and proliferation of cells during embryonic development and in maintaining tissue homeostasis [9].

Hh signaling is one of the most studied signaling pathways connected to the primary cilium, and it relies on the transport of receptors PATCHED1 (PTCH1), GPR161 and Smoothened (SMO) in and out of the cilium. During basal repression, PTCH1 and GPR161 are both localized in the cilium. The GLI proteins are phosphorylated by protein kinase A (PKA) and subsequently by CK1 and GSK3beta, leading to proteolytic cleavage to generate the repressor forms (GLI2R and GLI3R), preventing the transcription of target genes [10]. Binding of a ligand, e.g., Sonic Hh (SHh), to PTCH1, leads to the removal and degradation of PTCH1, allowing SMO to enter and accumulate in the ciliary membrane. The entrance of activated SMO subsequently directs removal of GPR161 [11]. SMO activation after transport into the cilium is required for activation of the pathway [10,12,13,14]. Activated SMO prevents the proteolytic cleavage of GLI. The full-length activated GLI proteins (predominantly GLI2A) translocate to the nucleus, turning on transcription of target genes, e.g., *GLI1* and *PTCH1* [6]. Chemical agonists and antagonists that directly bind to SMO lead to SMO accumulation in the primary cilium independently of SHh and PTCH1 [15].

As no protein synthesis takes place in the primary cilia, active transport of proteins into the primary cilium is required. This is carried out by intra-flagellar transport (IFT) to sustain primary cilium assembly, resorption and ciliary signaling [16]. IFT is a process that moves large complexes, termed IFT particles, along the axoneme of the primary cilium. These particles are composed of at least 30 proteins, organized in IFT-A and IFT-B subcomplexes. Early work suggested that the IFT-B complex was linked to a kinesin-II motor, Kif3, for anterograde transport towards the ciliary tip, whereas the IFT-A complex was connected to a dynein motor, Dync2h1, for retrograde transport towards the ciliary basis [17]. More recent investigations have shown that the two IFT complexes participate in ciliary transport in both directions [18,19]. BBS proteins play a critical role in regulating cilia composition. The BBSome complex, consisting of BBS1, BBS2, BBS4, BBS5, BBS7, BBS8 (TTC8), BBS9 (PTHB1) and BBS18 (BBIP1), plays a role in IFT, serving as an adaptor between cargo and the transport complex [20]. Another BBS protein complex, the chaperonin-complex, consisting of BBS6 (MKKS), BBS10 and BBS12, assembles the BBSome [21]. The role of the BBSome was previously suggested to be an adaptor for G-protein coupled receptors (GPCRs) in the process of ciliary delivery, but now studies suggest that the BBSome is mainly involved in retrograde transport and exit of GPCRs from the primary cilium, such as PTCH1, SMO and GRP161 [22,23,24,25,26,27].

Most quiescent cells in our bodies have primary cilia—this is also the reason why the symptom spectrum of BBS is so broad. The main affected retinal cell type in BBS is the photoreceptors [3,28]. Photoreceptors have a modified primary cilium connecting the inner segment, where all protein synthesis takes place, with the outer segment where the light transduction cascade takes place, making IFT fundamental for proper photoreceptor function [28,29]. Although it is mainly the photoreceptors that are affected in BBS, the retinal pigment epithelial (RPE) cells have gained interest throughout the last years. These cells have primary cilia and they carry out processes that are important for the function of the photoreceptor cells [30]. The RPE cells form a pigmented monolayer behind the retina between the photoreceptors and Bruchs membrane. This monolayer forms a polarized epithelium sheet with tight junctions and carry out several processes that support the function of the photoreceptor cells. These functions include exchange of nutrients, ions and metabolic waste between the photoreceptors and the blood stream. The RPE cells are able to absorb scattered light due to their pigmentation, they perform phagocytosis by shedding outer segments from the photoreceptor cells and perform the visual cycle where all-trans retinol is converted back to the active form, 11-cis retinal [30]. 

Ciliogenesis and Hh signaling have been reported to be disturbed in BBS [31]. Hh signaling defects have been associated with development of post-axial polydactyly during embryonic development of BBS-affected individuals [32,33,34]. Furthermore, BBS proteins and intact functional primary cilia have been proposed to be a prerequisite for RPE maturation [29,30,34,35]. Stem cells present an opportunity to create any cell type or cell system of interest, and after the ground-breaking discovery that somatic cells can be reversed to a pluripotent state, the interest in stem cell research has boomed [36,37,38]. Induced pluripotent stem cells (iPSC) have been used to study RPE cells, photoreceptor cells and retinal organoids [39,40].

In this study, ciliogenesis and Hh signaling were investigated in fibroblasts obtained from 5 patients with BBS due to pathogenic variants in *BBS1*, *BBS5* or *BBS10,* and in the hTERT-immortalized RPE cell line, (RPE-1) cells, in which the three different BBS genes, one at a time, were downregulated by small interfering RNA (siRNA) transfection. Furthermore, preliminary data demonstrating the ability of iPSC, reprogrammed from BBS1-fibroblasts, to differentiate into RPE cells are shown. 

## 2. Results

Various BBS genes have previously been shown to regulate cilia frequency and length with inconsistent conclusions [34,41,42]; therefore, we set out to determine the effect of the BBS1, BBS5 and BBS10 variants on ciliogenesis. To induce ciliogenesis, cells were grown at serum-reduced conditions for 48 h. Each cell forms at most one primary cilium.

### 2.1. Non-Functional BBS5 or BBS10 Leads to Reduced Ciliogenesis in Fibroblasts and RPE1 Cells

Fibroblast cell-lines obtained from one BBS patient with pathogenic variants in *BBS1* (named BBS1^−/−^), two patients (siblings) with pathogenic variants in *BBS5* (named BBS5^−/−^(A) and BBS5^−/−^(B), respectively) and two BBS patients with pathogenic variants in *BBS10* (named BBS10^−/−^(A) and BBS10^−/−^(B), respectively) (Table 1), and three anonymous control samples named Ctrl.A, Ctrl.D and Ctrl.E, were investigated. Investigation of the patient fibroblasts by immunofluorescence microscopy (IFM) with antibodies against the ciliary marker ARL13B showed that the cells were ciliated (Figure 1A). A detailed evaluation of the frequency of ciliated cells revealed significantly fewer BBS5^−/−^(B) cells with cilia (79.55%), but not BBS5^−/−^(A) cells (92.77%), compared to a pool of cells from the three control cell-lines (Ctrl.A, Ctrl.D and Ctrl.E) (92.60%). Also, the frequency of ciliated cells was significantly lower in BBS10^−/−^(A) (89.16%) and BBS10^−/−^(B) (85.78%) cells compared to the pool of the control cells (92.60%). However, also, a significant difference in number of ciliated cells was obtained between the three control cell-lines, and a significant reduction in frequency of ciliated cells was observed for Ctrl.D compared to Ctrl.A and Ctrl.E (Figure 1B). 

As the frequency of ciliated cells was reduced in BBS5^−/−^(B) cells but not in BBS5^−/−^(A) cells compared to the pool of control cells, and furthermore, the frequency of ciliated cells varied between the three control cell-lines, other genetic factors besides variants in the BBS genes might affect the ciliogenesis in these cells. To eliminate genetic differences not related to the BBS genes, we used siRNA transfection to knock-down BBS1, BBS5 and BBS10 from the human cell-line, hTERT-RPE1 (RPE1). Using this approach, we observed a profound and significant reduction in frequency of ciliated RPE1 cells after siRNA-mediated knockdown of BBS5 (60.20%) compared to RPE cells transfected with scramble siRNA sequence, siSCR (86.81%). Also, a significant reduction in frequency of ciliated cells was observed for siBBS10 RNA transfected RPE1 cells (76.40%), as shown in Figure 1C. These results indicate that functional BBS5 and BBS10 are important for ciliogenesis in fibroblasts and RPE1 cells, measured as frequency of ciliated cells, whereas no effect could be observed for BBS1.

### 2.2. Non-Functional BBS1 Leads to Reduced Cilia Length in Fibroblasts and RPE1 Cells

Investigation of the cilia length in patient fibroblasts revealed that the mean ciliary length was significantly shorter in BBS1^−/−^ cells (3.11 µm), whereas BBS5^−/−^ (4.31–4.14 µm) and BBS10^−/−^ (3.8–4.4 µm) all had significantly longer cilia compared to the pool of control cells (3.61 µm). There was no significant difference in the mean ciliary length between the three control cell-lines (Figure 1D). The ciliary lengths varied in all fibroblasts, but a tendency to broader length distribution in BBS5^−/−^(A), BBS5^−/−^(B) and BBS10^−/−^(A) fibroblasts was observed compared to control cells (Figure 1E). 

Investigation of the cilia length in siRNA transfected RPE1 cells verified an effect of BBS1 on the cilia length, as a significant reduction in cilia length was observed in RPE1 cells transfected with siBBS1 (3.07 µm), compared to scramble siRNA transfected control cells, siSCR (3.47 µm). However, a reduction in cilia length was also observed for RPE1 cells transfected with siBBS10 (2.85 µm). No effect on cilia length was observed after transfection with siBBS5 (Figure 1F). These results indicate that the effect of BBS1 on the cilia length is similar in fibroblasts and RPE1 cells, whereas the effects of the other BBS genes are more complex and might be cell-specific.

### 2.3. Impaired Canonical Hh Signaling in BBS1^−/−^, BBS5^−/−^ and BBS10^−/−^ Cells

It is well-established that a functional primary cilium is required for Hh signaling [43]. However, we have previously shown that an effect on the length of the primary cilia does not necessarily reflect an effect on the function [9]. To test the effect of BBS1, BBS5 and BBS10 on the function of the primary cilia, we investigated canonical Hh signaling in the patient fibroblast cells. We used the agonist purmorphamine (pur) to activate Hh signaling and the transcription of *GLI1* in response to Hh signaling as a marker for Hh activity. As shown (Figure 2A), pur-induced expression of *GLI1* was significantly reduced in all patient fibroblasts, compared to control fibroblasts (Ctrl). Almost no signal was detectable in the patient fibroblasts. Despite the very reduced signal, the *GLI1* expression in BBS1^−/−^ and BBS10^−/−^(A) fibroblasts was significantly increased as an effect of pur stimulation. 

### 2.4. Hh Was Also Dependent on BBS Genes in RPE1 Cells

The effects of BBS1, BBS5 and BBS10 on Hh signaling were also investigated in RPE1 cells transfected with siRNA against *BBS1*, *BBS5* or *BBS10*. As seen in Figure 2B, we observed a modest but still significantly increased pur-induced *GLI1* expression in RPE1 cells transfected with siBBS1 and siBBS10, whereas no significant increase was observed in RPE1 cells transfected with siBBS5. RPE1 cells transfected with scramble sequence (siSCR) displayed, as expected, a substantially significant pur-induced *GLI1* expression. Note that although siRNA transfection leads to a substantial reduction in expression of the BBS genes (Appendix A), the resulting BBS activity might explain the higher pur-induced *GLI1* expression in RPE1 cells compared to the BBS fibroblasts. These results indicate that Hh signaling is substantially reduced in BBS1^−/−^ and BBS10^−/−^ cells, and totally hampered in BBS5^−/−^ cells.

### 2.5. SMO Is Accumulated in Cilia of BBS1^−/−^, BBS5^−/−^ and BBS10^−/−^ Fibroblasts in the Absence of Hh Stimulation

SMO accumulation in cilia is usually seen after Hh stimulation [6], but loss of BBS proteins has previously been demonstrated to increase the basal level of SMO in cilia [31]. To test whether this was also the case in the present patient fibroblasts, we investigated the sub-cellular localization of SMO. As shown in Figure 2C, at the basal level, in the absence of pur, SMO could be detected in the cilia in all the BBS patient fibroblasts. In the control cells (Ctrl), SMO was only observed in a large number of cilia after pur stimulation (Appendix A). Investigation of cilia revealed that the fraction of cilia containing SMO, in the absence of pur stimulation, was significantly increased in all BBS fibroblasts (32–56%) compared to the pool of controls (Ctrl.A + Ctrl.D, 10.94%) (Figure 2D). Only after pur stimulation was a significant increase in ciliary SMO observed in the control cells, as expected (35.66%) (Figure 2D). The amount of SMO in the BBS cells in the absence of pur was similar to the amount of SMO in the control cells in the presence of pur. Except for BBS10^−/−^(A), no significant increase in SMO was observed in the BBS cell-lines as an effect of pur stimulation (Figure 2D). 

### 2.6. BBS Patient Fibroblasts Can Be Reprogammed into iPSCs

We have previously generated five iPSC clones (BBS^−/−^-iPSC) from the patient fibroblast cell-lines: BBS1^−/−^ iPSC-cl.10, BBS5^−/−^ iPSC-cl.3A, BBS5^−/−^ iPSC-cl.4A, BBS5^−/−^ iPSC-cl.5A and BBS10^−/−^ iPSC-cl.1A, using electroporation and non-integrating episomal plasmids [44,45,46]. To confirm that these iPSCs are in fact pluripotent despite expected aberrant Hh signaling, we tested the capacity of the iPSC cell-lines to differentiate into all three germ layers. Spontaneous differentiation was initiated by embryoid body formation, followed by adherent culture for a total of 21 days. The cells were subsequently analyzed by IFM. As previously published [44,45,46], all five iPSC cell-lines showed the capacity to differentiate into mesoderm, visualized by expression of α-smooth muscle actin (SMA), endoderm, visualized by expression of α-fetoprotein (AFP), and ectoderm, visualized by expression of βIII-tubulin (βTUB) (Figure 3). These results indicate that the BBS^−/−^-iPSC cell-lines were pluripotent and potentially capable of forming RPE cells as RPE cells are formed from the ectoderm germ layer.

### 2.7. BBS^−/−^ iPSCs Can Be Differentiated into RPE-Like Cells 

Although retinal degeneration in BBS is predominantly thought to be caused by functional and developmental abnormalities in retinal photoreceptors, abnormal RPE cells might also contribute to the degradation of the photoreceptors in BBS patients. It has previously been demonstrated that BBS8 and BBS14 (CEP290) are important for maturation of RPE cells [35]. We asked whether BBS variants, which lead to nonfunctional primary cilia, could affect the ability to generate RPE cells. To answer this question, we investigated the ability of the iPSC clone, BBS1^−/−^cl.10, to differentiate into RPE cells. For comparison, we included a control iPSC clone (K3 iPSC). Several protocols for differentiation of stem cells into retinal RPE cells have been described. The protocol developed by David Buchholz and Dennis Clegg was chosen for its simplicity, having cells in continuous adherent culture [47]. Modulation of WNT signaling has been shown to promote RPE differentiation, and addition of the canonical WNT pathway activator CHIR99021 at days 8, 10 and 12 is included [47,48,49]. After 14 days of differentiation, immature RPE precursor cells are formed. Culturing of these precursor cells were then continued, allowing the cells to mature during three passages (P0, P1, P2) of about 33 days each. 

To assess the maturation of the generated RPE cells at the differentiation endpoint (end of P2), morphology, pigmentation and expression of RPE-specific protein markers were investigated. At this point, the cells have been differentiating for a total of 110 days. As seen in Figure 4A, the iPSC-derived RPE cells have the expected hexagonal cobblestone-like shape and regular tight junctions, as judged by staining with antibody against ZO-1, a tight junction marker. No difference could be observed by comparison of the BBS1^−/−^ iPSC differentiated RPE cells and the K3 iPSC differentiated RPE cells. Both RPE cell types also formed primary cilia. Investigation of the cells by phase contrast and bright field reveal that the iPSC differentiated RPE cells yielded pigmented small compact cobblestone cells, however, in this case, a difference between the RPE cells was observed. The K3 iPSC differentiated RPE cells were more pigmented, compared to the BBS1^−/−^ iPSC differentiated RPE cells.

To assess the maturation process of the generated RPE cells, gene expression of *PAX6* (retinal precursor marker), *BEST1* and *RPE65* (both mature RPE markers) were investigated at different timepoints. mRNA was isolated at day 0 (iPSC), day 14 (d14) and at the end of each of the three subsequent passages: P0, P1 and P2 (Figure 4B). *PAX6* expression declined after day 14 of RPE differentiation, whereas *BEST1* and *RPE65* expression increased over time and peaked at P2. A similar pattern was obtained for differentiation of K3 iPSC and BBS1^−/−^ iPSC clones.

These results indicate that although the primary cilia in BBS1-defective cells are not working properly, leading to dampened Hh signaling, the BBS1^−/−^ iPSC cells were able to differentiate into at least RPE-like cells. The BBS1^−/−^ iPSC differentiated RPE cells express mature RPE markers at comparable levels with the K3 iPSC differentiated RPE cells. However, the reduced pigmentation of BBS1^−/−^ iPSC differentiated RPE cells might indicate that some of the characteristics of mature RPE cells are missing in BBS1^−/−^ iPSC differentiated RPE cells. 

## 3. Discussion

In the present study, we observed that BBS5 and BBS10 affect the number of cilia in the patient fibroblasts and the RPE1 cells, while this was not observed for BBS1. BBS5 seems to have a major effect on ciliogenesis compared to BBS10, however the reduced frequency of ciliated cells was only observed in BBS5^−/−^(B) patient fibroblasts but not in BBS5^−/−^(A) fibroblasts.

A reduction in cilia length was observed in both BBS1-defective fibroblasts and BBS1-reduced RPE1 cells. In contrast to BBS1, the effect of BBS5 and BBS10, according to effect on the cilia length, was not consistent for the two cell types. The effect of BBS10 on cilia length was the opposite in the two cell types, with increased cilia length in BBS10 patient fibroblasts and decreased cilia length in BBS10-reduced RPE1 cells. With respect to BBS5, an increased cilia length was observed in BBS5 patient fibroblasts, whereas no effect was observed in BBS5-reduced RPE1 cells. 

The observed discrepancies in effect of BBS proteins according to cilia frequency and cilia length may be due to cell- or species-specific differences, but also other genetic factors might contribute to the observed discrepancies. A large number of genes and signaling pathways have been demonstrated to affect cilia length [9,50,51]. Evidence for contribution of genetic factors, in our study, is supported by the observation that the frequency of ciliated cells was reduced in BBS5^−/−^(B) patient fibroblasts but not in BBS5^−/−^(A) fibroblasts compared to the pool of control cells. Furthermore, a significant difference between the cilia length in BBS10^−/−^(A) compared BBS10^−/−^(B) can be calculated (*p* = 6.41 × 10^−6^). The observed effect may furthermore depend on the specific BBS gene affected. Although BBS1 and BBS5 are both members of the BBSome, we observed different effects on both cilia frequency and ciliary length, indicating that even BBS proteins in the same complex may have different functions.

The effect of BBS genes on frequency and length of cilia has previously been investigated in several studies. Depletion of BBS5 or BBS8, but also depletion of BBS1, did not affect the cilia length in murine medullary collecting duct cells [41]. However, other studies demonstrated that BBS4- and BBS6-deficient murine renal medullary cells [42] and BBS6 and BBS8 murine knock-out (KO) RPE cells display shorter cilia and fewer ciliated cells [34]. Despite discrepancies between the obtained results, the overall picture indicated that the BBS proteins to some extent affect ciliogenesis. It has been shown that the BBSome is important for transporting tubulin in the primary cilium through IFT [42]. Thus, disturbing the BBSome may lead to abnormal trafficking of tubulin and altered ciliary length homeostasis [6,42]. Furthermore, Patnaik et al. found BBS proteins to protect ciliary length through interaction with Inversin, which is a regulator of Aurora A kinase that activates histone deacetylase 6 (HDAC6), which in turn destabilizes ciliary axonemal microtubules in RPE cells, leading to disassembly of the primary cilium [34].

To investigate the effect of BBS proteins on the function of the primary cilia, we took advantage of the tight connection between Hh signaling, the IFT and the primary cilium [52]. All five BBS fibroblast lines showed substantially reduced Hh signaling. Knock-down of *BBS1*, *BBS5* or *BBS10* in RPE1 cells gave similar results. BBS1, BBS5 and BBS10 were all important for Hh signaling. The observed accumulation of SMO in primary cilia, at basal conditions without Hh stimulation, in all five fibroblast cell-lines, might indicate that the accumulated SMO is inactive, explaining the aberrant Hh signaling. SMO accumulation in the primary cilium is a prerequisite but not sufficient for Hh signaling. SMO needs to be activated as well. Antagonists, such as cyclopamine, also provoke SMO accumulation in primary cilia, but without activation of Hh signaling [14,15]. Another explanation for the aberrant Hh signaling might be the reduced amount of activated GLI2, due to a lack of cilia export of GRP161. GPR161 catalyzes the guanosine diphosphate (GDP)-bound form of Gαs into the guanosine triphosphate (GTP)-bound form, leading to activation of adenyl cyclase (AC), high levels of cyclic adenosine monophosphate (cAMP) [23] and activation of protein kinase A (PKA), which phosphorylates GLI2 and GLI3, leading to their degradation or formation of repressor forms, GLI2Rep and GLI3Rep. The BBSome complex has been demonstrated to be important for export of GPR161 from cilia [25]. 

In the absence of Hh activation, SMO is normally ubiquitinated and removed from the primary cilia. The BBSome and BBS17 (LZTF1), BBS19 (IFT27) and IFT25 (an IFT-B complex protein) work together to remove SMO from the primary cilia. Also, BBS3 (Arl6), interacting with BBS1, regulates assembly of the BBSome and membrane attachment [51]. In the presence of Hh activation, the same proteins work together to remove ubiquitinated PTCH1 from the primary cilia [52]. In the lack of a functional BBSome-BBS17-BBS19-IFT25 transport system, it is likely that both PTCH1, SMO and GPR161 accumulate in the primary cilia because none of the proteins can be removed. SMO (and PTCH1) undergoes lateral transport in and out of primary cilia at steady-state, and if export is hampered, this might lead to accumulation in the primary cilia [25,31,53,54,55,56,57,58,59]. BBS1 and BBS5 are both members of the BBSome, whereas BBS10 is a member of the BBS-chaperonin complex. 

The observed lack or substantial reduced Hh activity, as a result of pathogenic variants in *BBS1*, *BBS5* and *BBS10,* might be explained by the importance of the BBSome for trafficking of receptors important for Hh signaling. In contrast, it was previously found that only depletion of BBS1 and not depletion of BBS4, BBS5 or BBS8 (all members of the BBS-chaperonin complex), impairs ciliary trafficking of PC1 in kidney epithelial cells [41]. It was suggested, based on this observation, that the different BBS proteins in the BBSome may possess protein-specific functions [41]. It is also not certain that all BBS genes are equally important for Hh signaling. This is supported by several observations. In murine fibroblasts, lack of BBS7 only led to a 20–30% decrease in Hh activation [31], a very small reduction compared to our observations. It might indicate that BBS7 has a less important function regarding Hh signaling. Furthermore, normal Hh signaling was observed in patient fibroblasts with pathogenic BBS2 variants [60] and even increased Hh signaling was observed in patient fibroblasts with pathogenic variants in BBS14 (CEP290) [61]. 

Interestingly, the transcription factor NRF2 has been demonstrated by Liu et al. [62] to negatively regulate primary ciliogenesis and Hh signaling, and this was at least partly due to blockage of cilia entry of BBS4, due to upregulation of the ubiquitin binding protein p62, resulting in sequestration of BBS4 into inclusion bodies. Thus, upregulation of NRF2 must be expected to lead to a similar effect as depletion of BBS genes. In fact, it was found that upregulation of NRF2 leads to reduced percentage of ciliated cells and reduced Hh signaling [62]. However, in contrast to our observation, upregulation of NRF2 was found to hamper the ciliary entrance of SMO, probably by NRF2-mediated upregulation of PTCH1. As Liu et al. [62] used SHh as a ligand, whereas we used pur, which interact directly with SMO bypassing PATCH, the results are not directly comparable. NRF2 was found to be upregulated by the Hh inhibitor-4 (HPI-4), supporting the negative regulation of NRF2 on Hh signaling [62]. Further studies are needed to pinpoint the exact role of the BBS proteins in Hh signaling.

We found that iPSC generated by reprogramming of BBS1^−/−^, BBS5^−/−^ and BBS10^−/−^ fibroblasts had the capacity to differentiate into all three germ layers, including ectoderm, which is a prerequisite for the ability to differentiate into RPE cells. Although it is mainly the photoreceptors that are affected in BBS, we speculated that the RPE cells might be affected as well. As BBS1^−/−^ iPSC differentiated RPE were less pigmented compared to K3-iPSC differentiated RPE cells, a difference in maturity might be present. Defective RPE cells might even precede photoreceptor degeneration, as May-Simera et al. have shown that the RPE cells were affected before the photoreceptor cells in ciliopathy mice: BBS8 KO mice [35]. They found that BBS8 KO cells display ZO-1 irregular immunostaining and increased canonical WNT signaling. Also, investigation of RPE cells differentiated from patient-specific iPSCs from a patient carrying pathogenic variants in BBS14 (CEP290) [35] verified the importance of the primary cilium. They demonstrate that the primary cilium is essential for RPE maturation, because it suppresses canonical WNT signaling, leading to cell cycle exit and activation of PKCδ pathways. They observed that treatment of iPSC-RPE with WNT inhibitors led to higher expression of the maturation marker RPE65. For RPE differentiation, we used a protocol where a WNT activator was added during the first 14 days. Using this protocol, we were able to obtain RPE65 expressing RPE cells with regular ZO-1 staining from both K3-iPSC and BBS1^−/−^ iPSC. We found that BBS1^−/−^ iPSC differentiated RPE were less pigmented compared to K3-iPSC differentiated RPE cells, but otherwise, no obvious difference was observed. As BBS8 interact directly with the canonical WNT effectors Inversion and Jouberin, the lack of BBS8 might be more devastating compared to the loss of BBS1. Further investigation of the RPE cells is needed, including phagocytosis activity, to clarify the maturity of the BBS1^−/−^ iPSC differentiated RPE cells versus the K3-iPSC differentiated RPE cells. Future investigations might also include the effect of other BBS genes in addition to *BBS1*, and test the effect of WNT inhibitors on RPE cell maturation. 

## 4. Materials and Methods 

### 4.1. Cell Culture

Fibroblast cells from five BBS patients were obtained through skin biopsy and cells from three healthy control individuals were included for comparison (Control A, Control D, Control E). The BBS1 patient was compound heterozygous for two missense mutations in *BBS1*, the two BBS5 patients were homozygous for a missense mutation in *BBS5* and the two BBS10 patients were homozygous for a frameshift mutation in *BBS10* (Table 1). RPE1 (hTERT-RPE1) cells were a kind gift from Lotte Bang Pedersen (University of Copenhagen, Department of Biology, Section for Cell Biology and Physiology), originally obtained from ATCC (hTERT RPE-1 ATCC ® CRL-4000TM). All cells were grown in either standard growth conditions, DMEM-F12 + GlutaMAX (Gibco), 10% fetal bovine serum (FBS) (Gibco, Thermo Fisher Scientific, Waltham, MA, USA) and 1% penicillin-streptomycin (pen/strep) (Gibco), or serum-reduced growth conditions with FBS reduced to 0.5%.

### 4.2. Gene Knockdown with siRNA

Cells were incubated with 25 nM negative control siRNA (siSCR), siBBS1 (SI03191622, Qiagen), siBBS5 (SI04339020, Qiagen, Germantown, MD, USA) and siBBS10 (SI04270371, Qiagen) overnight using DharmaFECT 1 (Dharmacon, Lafayette, Indiana, CO, USA) as per the manufacturer’s instructions. The next morning, medium was changed to serum-reduced (0.5% FCS) for Hh stimulation. Knock-down efficiency was assessed using quantitative Polymerase Chain Reaction (qPCR). For ciliogenesis and/or Hh stimulation, the medium was changed to serum-reduced (0.5% FCS) medium one day after transfection. 

### 4.3. Ciliogenesis and Hh Stimulation

To induce ciliogenesis, the cells were grown in serum-reduced (0.5% FCS) medium for 48 h. To induce Hh stimulation, cells were stimulated with 5 μM pur for the last 24 h.

### 4.4. qPCR

RNA was purified using the GeneJET RNA purification kit (thermo fisher scientific) and cDNA was synthesized using the high-capacity cDNA kit (Thermo Fisher Scientific, Warrington, UK). Taqman probes against endogenous *GAPDH* (Hs99999905_m1), *GLI1* (Hs01110766_m1), BBS1 (Hs00226452_m1), BBS5 (Hs00537098_m1), BBS10 (Hs00379769_g1), *PAX6* (Hs00240871_m1), *BEST1* (Hs04397293_m1) and *RPE65* (Hs01071462_m1), all from Thermo Fisher Scientific, were used in combination with TaqMan™ Universal PCR Master Mix (Thermo Fisher Scientific, Warrington, UK) on the 7500-fast system (Applied Biosystems, Warrington, UK). Three technical replicates were tested per sample. The relative standard curve method was used, and samples were normalized to *GAPDH* and a control sample. For the RPE cells, the ΔΔCT method was applied, and values were normalized to *TBP* and the control-RPE at day 0 (iPSC).

### 4.5. Immunofluorescence Microscopy

To induce primary cilia formation, fibroblasts were grown in serum-reduced media for 48 h. The RPE cells were grown on Matrigel-coated Lab-Tek 8-well chamber slides (Thermo Fisher Scientific #177402). The cells were subsequently fixed with 4% paraformaldehyde (Hounisen, Midtjylland, Denmark) for 15 min, permeabilized with 0.2% Triton X-10 in PBS and blocked with 3% bovine serum albumin (BSA) in permeabilization buffer. Incubation with primary antibodies diluted in 3% BSA was done for 45 min at room temperature or overnight at 4 °C. Secondary antibody incubation was carried out for 1 h at room temperature before mounting. Nuclei were visualized using DAPI. All images were obtained using an Olympus Fluoview 1000 confocal microscope and images were analyzed using ImageJ. The following antibodies were used: Anti-α-Acetylated tubulin (Sigma-Aldrich #T6793, St. Louis, MO, USA), Anti-Smoothened (Abcam #ab38686, Cambridge, UK), Anti-ARL13B (Proteintech #1711-1-AP, Rosemont, IL, USA), anti-α-smooth muscle actin (SMA) (Dako #M0851), anti-α-fetoprotein (AFP) (Dako #A0008), anti-βIII tubulin (βtub) (Sigma-Aldrich #T8660) and anti-ZO-1 (Thermo Fisher Scientific #40-2200). 

### 4.6. Statistical Methods

Student’s *t*-test or the χ^2^-test were used as indicated. The *t*-test was calculated as an un-paired, one-tailed or two-tailed test, as stated. For both methods, the significance level at 0.05 was chosen. Significance starts are in the following categories: * *p* ≤ 0.05, ** *p* ≤ 0.005, *** *p* ≤ 0.0005. Only data from three separate biological experiments were used for statistical analysis.

### 4.7. iPSC Generation and Culture

All iPSCs were cultured on hESC-qualified Corning^®^ Matrigel^®^ (Corning #354277, Corning, NY, USA) in mTeSRTM1 (STEMCELL Technologies #85850, Vancouver, BC, Canada). Reprogramming of BBS1-iPSC is described in Reference [44], BBS5 iPSC in Reference [46] and BBS10-iPSC in Reference [45].

### 4.8. In Vitro Differentiation

The iPSCs were treated with 0.5 mM ultrapure EDTA (Gibco) and plated in ultra-low adhesion plates (CORNING) in mTeSR1 media with ROCK inhibitor, to induce formation of embryoid bodies. On day two, media was changed to DMEM-F12 + GlutaMAX (Gibco), 20% knock-out serum replacement (Gibco, Thermo Fisher Scientific, Denmark)), 1x non-essential amino acids (Sigma), 0.1 mM 2-mecaptoethanol (Sigma-Aldrich, Merck, Germany) and 1% pen/strep. After one week of suspension culture, the aggregates were transferred to adherent culture in DMEM F-12 with GlutaMAX, 10% FBS and 1% pen/strep media on gelatin (Sigma)-coated coverslips. Morphological changes were observed, and after two weeks of adherent culture, the cells were fixed and investigated by immunocytochemistry.

### 4.9. RPE Differentiation

The protocol developed by Clegg [47,48,49] was applied. Briefly, iPSCs were cultured with different small molecules and growth factors for 14 days and then left to mature for three passages (P0, P1 and P2) of 28–30 days in continuous adherent 2D culture. The combination of growth factors added and small molecules during the initial 14 days are as follows:Day 0 and day 1: 10 mM NIC, 50 ng/mL noggin, 10 ng/mL DKK-1, 10 ng/mL IGF-1.Day 2: 10 mM NIC, 5 ng/µL FGF-basic, 10 ng/mL noggin, 10 ng/mL DKK-1, 10 ng/mL IGF-1.Day 4: 100 ng/mL activin A, 10 ng/mL DKK-1, 10 ng/mL IGF-1.Day 6: 100 ng/mL activin A, 10 µM SU 5402.Days 8, 10, 12: 100 ng/mL activin A, 10 µM SU 5402, 3 µM CHIR99021.

After the initial 14 days, the iPSC-RPE cells were cultured in X-vivo 10 on growth factor-reduced Matrigel-coated surfaces. At the end of the initial 14 days, and at the end of all three passages, P0, P1 and P2, a fraction of the cells was harvested for RNA studies and immunofluorescence microscopy. 

## 5. Conclusions

BBS1, BBS5 and BBS10 affect ciliogenesis, and Hh signaling was substantially impaired/absent, indicating defective primary cilia. BBS1-defective iPSC could be differentiated into RPE65 expressing less pigmented RPE-like cells.

## Figures and Tables

**Figure 1 ijms-22-01345-f001:**
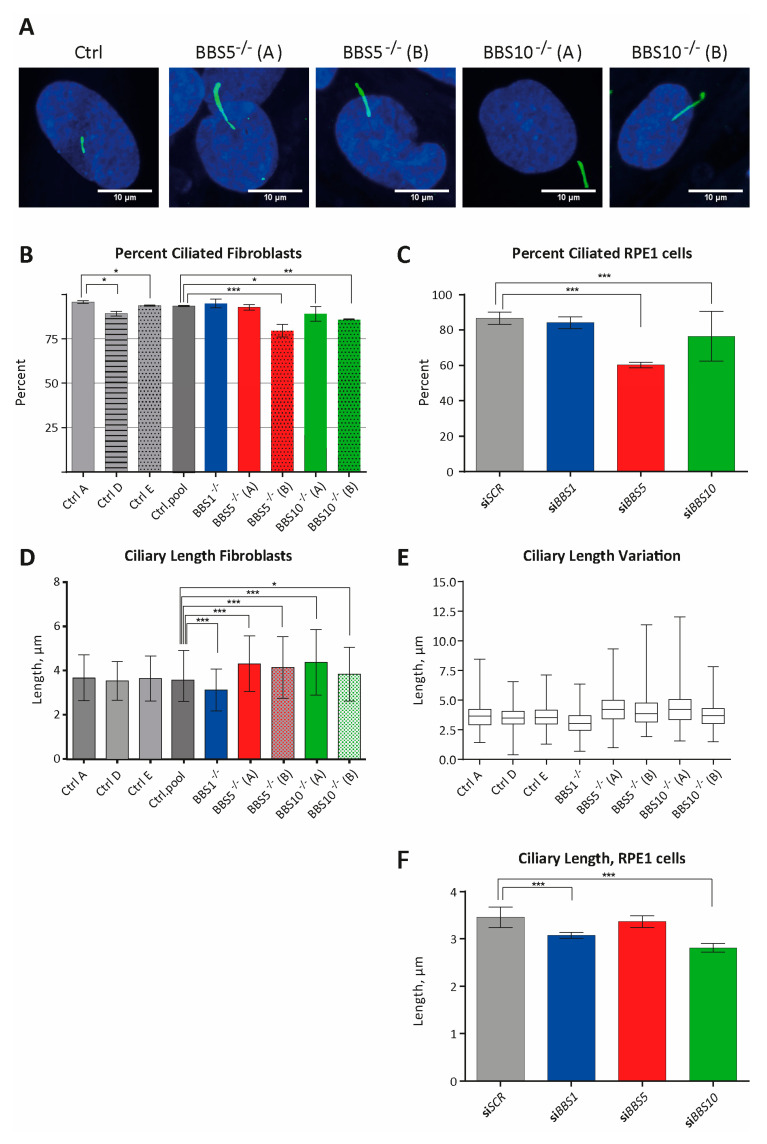
Investigation of the primary cilium in fibroblast obtained from patients with bardet biedl syndrome (BBS) and in the hTERT-immortalized retinal pigment epithelial cell-line, (RPE-1) transfected with small interfering RNA (siRNA) against the BBS genes. (**A**) IFM analysis of primary cilia. Primary cilia were labeled with anti-ARL13B antibody (green). Nuclei were visualized with DAPI staining (blue). Controls (Ctrl, here CtrlD) are also shown. Scale bars 10 µM. (**B**) Percentage of patient fibroblasts with cilia. Compared to percentage of ciliated cells in a pool of control fibroblasts (Ctrl.pool = CtrL.A + Ctrl.D + Ctrl.E; 92.60%, *n* = 798 cells), the percentage of ciliated cells was significantly lower in BBS5^−/−^(B) (79.54%, *n* = 308 cells, *** *p* =5.13 × 10^−10^), BBS10^−/−^(A) (89.16%, *n* = 369 cells, * *p* = 0.0494) and BBS10^−/−^(B) (85.78%, *n* = 225 cells, ** *p* = 0.0015) fibroblasts. No significant differences were observed for BBS1^−/−^ (94.98%, *n* = 239, *p* = 0.202) and BBS5^−/−^(A) (92.77%, *n* = 249, *p* = 0.9301) compared to the pool of control fibroblasts. There was a significant difference in percentage of ciliated cells between the three controls (Ctrl). The percentage was significantly lower in Ctrl.D (89.18%, *n* = 305 cells) compared with Ctrl.A (95.72%, *n* = 234 cells; Ctrl.D/Ctrl.A: * *p* = 0.0054) and Ctrl.E (93.82%, *n* = 259 cells; Ctrl.D/Ctrl.E: * *p* = 0.037). No significant difference was observed between CtrlA and CtrlE (CtrlA/CtrlE: *p* = 0.34). (**C**) Percentage of siBBS RNA-treated RPE1 cells with cilia. The cilia were labeled with anti ARL13B antibody and nuclei were visualized with DAPI staining. For siRNA efficacy, see Appendix A. Compared to percentage of ciliated cells in siSCR transfected RPE1 cells (86.81%, *n* = 379 cells), the percentage of ciliated cells were significantly decreased in RPE1 cells transfected with siBBS5 (60.20%, *n* = 294 cells, *** *p* = 2.2802 × 10^−15^) and siBBS10 (76.40%, *n* = 322 cells, *** *p* = 0.000347). No significant differences compared to siSCR transfected cells were obtained for siBBS1 (84.49%, *n* = 361 cells, *p* = 0.36789). (**D**) Quantification of primary cilia length in patient fibroblasts. No significant difference was obtained by comparing the three control fibroblasts lines (Ctrl.A, *n* = 224 cells; Ctrl.D, *n* = 270 cells; Ctrl.E *n* = 243 cells. Ctrl.A/Ctrl.D: *p* = 0.0918; Ctrl.A/Ctrl.E: *p* = 0.675; Ctrl.D/Ctrl.E: *p* = 0.207). A pool of all the controls (*n* = 737 cells) was used for comparison with the five BBS patient fibroblasts lines. Compared to the controls, BBS1^−/−^ (*n* = 222, *p* = 3.77 × 10^−11^) had significantly shorter cilia, whereas BBS5^−/−^(A) (*n* = 228, *** *p* = 1.83 × 10^−13^), BBS5^−/−^(B) (*n* = 242, *** *p* = 8.77 × 10^−8^), BBS10^−/−^(A) (*n* = 333, *** *p* = 2.00×10^−16^) and BBS10^−/−^(B) (*n* = 202, * *p* = 0.0165) all had significantly longer cilia. (**E**) Cilia length variation visualized as a boxplot of (C) showing the length variation for all cell-lines. Compared to control cell-lines, BBS5^−/−^(A), BBS5^−/−^(B) and BBS10^−/−^(A) had broader length distributions. (**F**) Quantification of primary cilia length in siBBS-treated RPE1 cells. The cilia length was significant shorter in RPE1 cells treated with siBBS1 (*n* = 361, *** *p* = 2.29 × 10^−8^), and siBBS10 (*n* = 322, *** *p* = 3.54 × 10^−19^) compared to siSCR-treated control RPE cells, whereas no significant difference was observed between siBBS5- (*n* = 294, *p* = 0.21) and siSCR-treated cells (*n* = 329). SiBBS transfection led to a substantial reduction in expression of the BBS genes (Appendix A). In (**B**,**C**), the number of cells (*n*) investigated consist of pooled data from three separate experiments and significance was determined using the χ^2^-test. In (**D**,**F**), the number of cilia (*n*) measured consist of pooled data from three separate experiments. *p*-values and significance were determined using Student’s *t*-test, two-tailed. Significance levels *p* < 0.05, * *p* < 0.05, and *** *p* < 0.0005 were used. Error bars represent the standard deviation.

**Figure 2 ijms-22-01345-f002:**
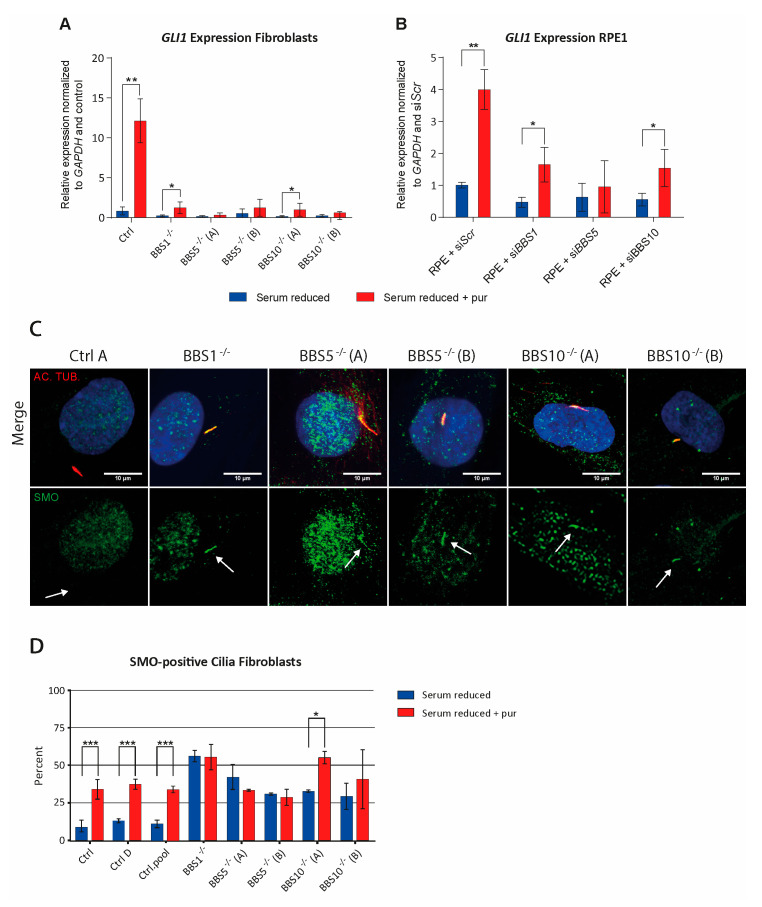
Investigation of Hedgehog (Hh) signaling in control and BBS-defective cells. (**A**) Investigation of Hh signaling fibroblast. Expression profile of *GLI1* mRNA was normalized to the amount of endogenous *GADPH* mRNA. Pur-induced expression of *GLI1* was significantly increased in control cells compared to patient fibroblasts (BBS1^−/−^: * *p* = 0.015711, BBS5A^−/−^: * *p* = 0.012758, BBS5B^−/−^: * *p* = 0.014811, BBS10A^−/−^: * *p* = 0.014944, BBS10B^−/−^: * *p* = 0.013701). Compared to unstimulated cells, a significant increase in *GLI1* expression as an effect of pur stimulation was observed in the control (Ctrl), BBS1^−/−^ and BBS10A^−/−^ fibroblasts (Ctrl: ** *p* = 0.001787, BBS1^−/−^: * *p* = 0.047977, BBS10A^−/−^ = * *p* = 0.039275) but not in the other cell types (BBS5A^−/−^: *p* = 0.148291, BBS5B^−/−^: *p* = 0.232452, BBS10B^−/−^: *p* = 0.098018). (**B**) Investigation of Hh signaling in siRNA transfected RPE1 cells. Expression profile of *GLI1* was normalized to the amount of endogenous *GADPH*. Cells transfected with siSCR showed a significant increase in *GLI1* expression after pur stimulation (siSCR * *p* = 0.038869). The increase in siBBS1 and siBBS10 transfected cells was smaller, but still significant, compared to unstimulated cells (siBBS1: * *p* = 0.020942, siBBS10: * *p* = 0.046011). No significant effect was observed in siBBS5 transfected cells (siBBS5: *p* = 0.323142). SiRNA transfection led to a substantial reduction in expression of the BBS genes (Appendix A). (**C**) Ciliary localization of SMO in fibroblast lines. The cells were labeled with anti-AC-TUB (cilia marker, red) and anti-SMO antibody (green). Nuclei were visualized with DAPI staining (blue). Scale bars 10 µM. Arrows indicate ciliary localization. In contrast to control fibroblasts, SMO was observed in a large number of cilia in all the BBS patient fibroblasts. In the control cells, SMO was only observed in a large number of cilia after pur stimulation (Appendix A). (**D**) Quantification of cilia with SMO. The fibroblasts were grown under serum-reduced conditions (0.05% FCS) for 48 h in the presence or absence of 5 µM pur for the final 24 h, as indicated. SMO was present in a large number of the cilia in all BBS patient fibroblasts, both in the presence and the absence of pur. Significantly more SMO was present in the BBS fibroblast cells compared to a pool of control cells (Ctrl.pool: Ctrl.A + Ctrl.D, *n =* 315) in the absence of pur (BBS1^−/−^: *** *p* = 1.28 × 10^−12^, *n* = 156, BBS5A^−/−^: *** *p* = 2.76 × 10^−8^, *n* = 145, BBS5B^−/−^: *** *p* = 6.95 × 10^−5^, *n* = 136, BBS10A^−/−^: *** *p* = 2.26 × 10^−5^, *n* = 134, and BBS10B^−/−^: *** *p* = 0.00014, *n* = 141). The control cells had a significant increase in ciliary SMO localization after pur stimulation (*** *p* = 4.58 × 10^−9^, *n* = 349/426). All BBS fibroblasts showed a significantly higher basal level of ciliary SMO that only increased significantly in BBS10A after pur stimulation (BBS1^−/−^: *p* = 0.960, *n* = 222, BBS5A^−/−^: *p* = 0.371, *n* = 228, BBS5B^−/−^: *p* = 0.802, *n* = 242, BBS10A^−/−^: * *p* = 0.045, *n* = 333, BBS10B^−/−^: *p* = 0.219, *n* = 202). All included data were pooled from three independent experiments (*n* = 3). In (**A**,**B**), Student’s *t*-test, one-tailed, with *p* < 0.05 significance level was performed. In (**D**), the χ^2^-test was performed with significance levels * *p* < 0.05, and *** *p* < 0.0005. Error bars represent the standard deviation.

**Figure 3 ijms-22-01345-f003:**
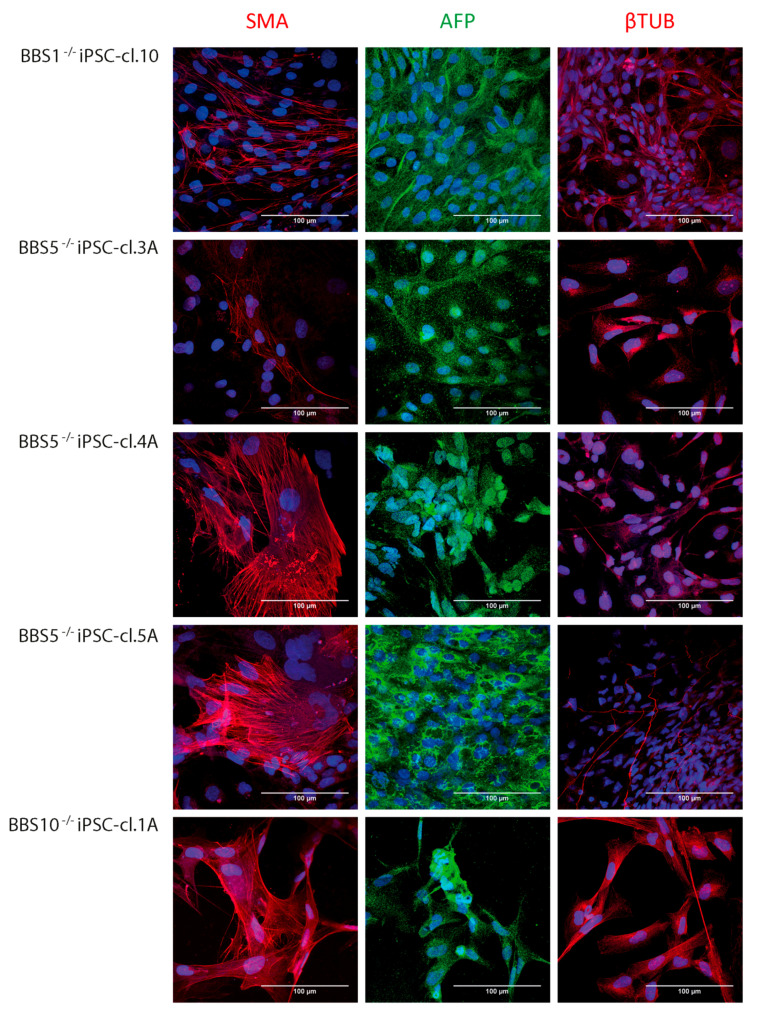
Immuno-fluorescence microscopy (IFM) analysis of BBS-induced pluripotent stem cells (iPSCs) after spontaneous differentiation towards the three germ layers. IFM performed after three weeks of spontaneous differentiation. The cells were labelled with anti-α-smooth muscle actin (SMA; mesoderm marker), anti-α-fetoprotein (AFP; endodermal marker) and anti-βIII-tubulin (βTUB; ectoderm marker). All the 5 different iPSC clones (BBS1^−/−^ iPSC-cl.10, BBS5^−/−^ iPSC-cl.3A, BBS5^−/−^ iPSC-cl.4A, BBS5^−/−^ iPSC-cl.5A and BBS10^−/−^ iPSC-cl.l1A) generated from BBS fibroblast were able to spontaneously differentiate into cells of all three germ layers (scalebar 100 µm).

**Figure 4 ijms-22-01345-f004:**
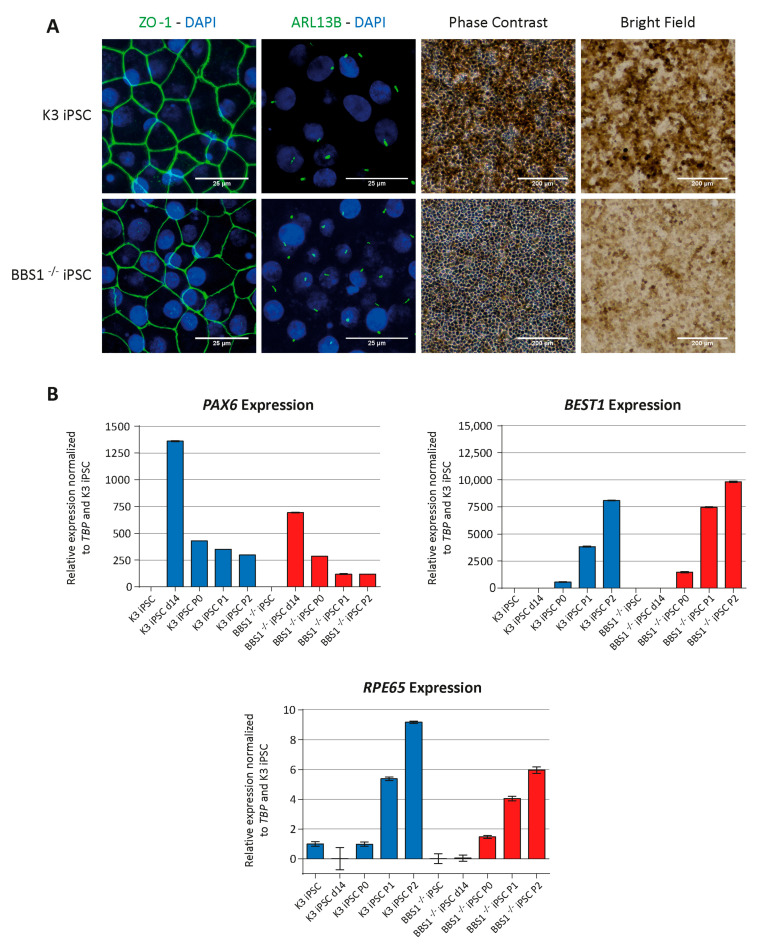
Differentiation of BBS1-iPSC into RPE-like cells. Control K3 iPSC and BBS1^−/−^ iPCS were analyzed for RPE phenotype, after differentiation for 110 days (end of P2). (**A**) IFM after incubation with antibodies against ZO-1 (tight junction marker) and ARL13B (Cilia marker). Nuclei were visualized with 4′,6-diamidino-2-phenylindole (DAPI) staining (Panels 1 and 2), scale bar 25 µM. Phase contrast pictures (Panel 3), scale bar 200 µM. Bright-field images (Panel 4), scale bar 200 µM. Both control K3 iPSC and BBS1^−/−^ iPCS were able to form tight junctions (Panel 1) and primary cilia (Panel 2). The cells had obtained the characteristic cobblestone morphology of RPE cells (Panel 3). Both cell-lines became pigmented (Panel 4). (**B**) Gene Expression. mRNA was isolated at different time points during the differentiation process and quantitative Polymerase Chain Reaction (qPCR) was performed. mRNA was isolated at day 0 (iPSC clone), day 14 after initiation of differentiation (d14) and at the end of the three passages, P0, P1 and P2. Expression profile of *PAX6* (retinal precursor marker), *BEST1* (mature RPE marker) and *RPE65* (mature RPE marker) was normalized to the expression of the endogenous *TBP* gene encoding the TATA-box binding protein. Data derived from one experiment (technical triplicates); no statistics performed.

**Table 1 ijms-22-01345-t001:** Variant information on the used patient cell-lines.

Patient Number (Abbreviation)	Variant
Patient 1 (BBS1)	Compound heterozygous: BBS1 c.1169T>G, p. (Met390Arg)/c.1135G>C, p.(Gly370Arg)
Patient 2 (BBS5A)	Homozygous: BBS5 c.214G>A, p.(Gly72Ser)
Patient 3 (BBS5B)	Homozygous: BBS5 c.214G>A, p.(Gly72Ser
Patient 4 (BBS10A)	Homozygous: BBS10 c.271insT (p. Cys91fs*95)
Patient 5 (BBS10B)	Homozygous: BBS10 c.271insT (p. Cys91fs*95)

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
