# Peer review of "BBS Proteins Affect Ciliogenesis and Are Essential for Hedgehog Signaling, but Not for Formation of iPSC-Derived RPE-65 Expressing RPE-Like Cells"

_ijms, 2021, doi:10.3390/ijms22031345_

Round 1
Reviewer 1 Report
The authors describe investigations of the rare, inherited disease, the Bardet-Biedl syndrome (BBS). The authors have fibroblasts from 3 patients who have mutated BBS-1, -5 or -10 genes, respectively and have constructed 3 iPSC cell lines where the corresponding BBS genes were knocked down via siRNA. The authors investigate cilia numbers and length in these cells compared to control cells as well as smo expression in the ciliae. The researchers identify reduced ciliae numbers and length in the affected cells compared to control. Smo is highly abundant within the ciliae without signs of activation of the Hedgehog pathway.
Comments:
1) There were discrepancies in the length of ciliae between the BBS5 and BBS10 fibroblasts and PRE cells. The authors should comment on this difference. Was it only due to the low number of patients involved?
2) SMO was abundant in the ciliae of the mutated cells. But the GLI1 expression was low indicating an inactivity of the hedgehog pathway. The authors should speculate on the mechanism, e.g. discussing a possible effect of NRF2 or HPI-4.
Author Response
We would like to thank the reviewer for the important question.
We have now improved the discussion according to the discrepancies in the length between the BBS5 and BBS10 fibroblasts and RPE cells (line 333-351)
and added discussion of NRF2 (line 410-413)
Reviewer 2 Report
In this manuscript there are two main goals the researchers want to follow. The first one is to evaluate the cellular phenotype of fibroblast derived for BBS1, BBS5 and BBS10 patients. The second is to create iPSC cells and derive them in RPE like cells.
First the authors test ciliogenesis in the primary cells lines and compare it to siRNA KD of the three genes in RPE1 cells. This work and the results are not really novel and it shows very subtle differences in cilia formation compare with wild-type controls. It is surprising the differences observed between on type of KD (fibroblast) and siRNA depleted cells. The authors discuss the possible reasons behind these discrepancies. This is followed by the analysis of the SHH pathway in the BBS depleted cell lines, that show accumulation of SMO in the cilia
Then the authors instead of looking in detail on what is happening if their BBS depleted cells with the SHH pathway, decide to test if the can create iPSC and reprogramme BBS1 null cells into RPE cells. Results seem to suggest they have achieve their goal and the cells express 3 RPE markers. Unfortunately, no more work is presented, and including the authors admit further work is necessary.
Author Response
We would like to thank the reviewer for the comments